# Theoretical Studies of Leu-Pro-Arg-Asp-Ala Pentapeptide (LPRDA) Binding to Sortase A of *Staphylococcus aureus*

**DOI:** 10.3390/molecules27238182

**Published:** 2022-11-24

**Authors:** Dmitry A. Shulga, Konstantin V. Kudryavtsev

**Affiliations:** 1Department of Chemistry, Lomonosov Moscow State University, Leninskie Gory 1/3, 119991 Moscow, Russia; 2Laboratory of Molecular Pharmacology, Pirogov Russian National Research Medical University, Ostrovityanova Street 1, 117997 Moscow, Russia

**Keywords:** *Staphylococcus aureus*, anti-virulence drugs, binding mode, molecular docking, molecular dynamics, binding mode analysis

## Abstract

Sortase A (SrtA) of *Staphylococcus aureus* is a well-defined molecular target to combat the virulence of these clinically important bacteria. However up to now no efficient drugs or even clinical candidates are known, hence the search for such drugs is still relevant and necessary. SrtA is a complex target, so many straight-forward techniques for modeling using the structure-based drug design (SBDD) fail to produce the results they used to bring for other, simpler, targets. In this work we conduct theoretical studies of the binding/activity of Leu-Pro-Arg-Asp-Ala (LPRDA) polypeptide, which was recently shown to possess antivirulence activity against *S. aureus*. Our investigation was aimed at establishing a framework for the estimation of the key interactions and subsequent modification of LPRDA, targeted at non-peptide molecules, with better drug-like properties than the original polypeptide. Firstly, the available PDB structures are critically analyzed and the criteria to evaluate the quality of the ligand–SrtA complex geometry are proposed. Secondly, the docking protocol was investigated to establish its applicability to the LPRDA–SrtA complex prediction. Thirdly, the molecular dynamics studies were carried out to refine the geometries and estimate the stability of the complexes, predicted by docking. The main finding is that the previously reported partially chaotic movement of the β6/β7 and β7/β8 loops of SrtA (being the intrinsically disordered parts related to the SrtA binding site) is exaggerated when SrtA is complexed with LPRDA, which in turn reveals all the signs of the flexible and structurally disordered molecule. As a result, a wealth of plausible LPRDA–SrtA complex conformations are hard to distinguish using simple modeling means, such as docking. The use of more elaborate modeling approaches may help to model the system reliably but at the cost of computational efficiency.

## 1. Introduction

Multidrug-resistant infections are a major cause of hospital mortality [1,2]. The main root cause is believed to be the widespread and uncontrolled use of traditional antibiotics, which are aimed at intervening in one of the signaling pathways directly related to bacteria survival. The latter causes evolutionary pressure, which results in mutations helping to avoid susceptibility to the specific antibiotic. The rate at which bacteria are capable of developing resistance far exceeds the rate at which humans are able to produce new antibiotics, so new approaches should be tried. One of the prospective directions [2,3] is to target the virulence and not the survival of bacteria per se. This way, evolutionary pressure should be lower. Hopefully, antivirulence drugs will help to alleviate the explosive growth of the bacteria population inside the human host, which is the main reason for both sickness conditions and complications, without trying to kill bacteria. In that case, the immune system should eventually cope with bacteria.

Sortase A (SrtA) of *Staphylococcus aureus* has long been validated as a relevant target to combat virulence of bacteria [1,4,5]. SrtA is a transpeptidase that functions on the outer side of the bacteria cell and participates in displaying surface proteins to the outer cell wall, which are crucial for bacterial virulence. The specific substrates for transpeptidation are recognized by the presence of the LPxTG sequence, called a “sorting sequence”, in a polypeptide chain of substrates [1].

Despite being well validated as a promising target, SrtA appeared to be a hard target for designing new drugs. To date, several experimental structures, fitting the requirements for structure-based drug discovery (SBDD) are available in the PDB. Still, despite numerous efforts, no drug candidates, or even low nanomolar leads, are known currently.

There are two main sources of difficulties which are related to the structure of SrtA and which may help to explain the current state. Firstly, the major part of the binding site of SrtA is the bottom of the flat and shallow pocket formed by the β4, β7 and β8 sheets of the rigid β-barrel core of the enzyme. Mainly the purely hydrophobic side chains are represented in the β-sheets of the site’s bottom. It is hard to form both affine and specific interactions in such conditions. The specific directed interactions could be established with the sides of the binding site, but the second difficulty is that the sides, represented in particular with the β6/β7 and β7/β8 loops, undergo rather complex motions with different time-scale dynamics [6,7,8]. The latter results in the binding site looking significantly different even in PDB structures, not to mention the possible behavior in solution. Since the above difficulties complicate the straight-forward SBDD approach, any source of experimental information should be used to reconstruct the requirements of the binding site to a ligand [9]. In particular, starting from a validated hit compound becomes even more valuable than it is in the regular drug discovery project.

Several studies have been devoted to the development of both covalent and non-covalent small molecule inhibitors of SrtA [2,4,10,11,12,13]. None of those inhibitors showed activity in the low nanomolar range, being a common prerequisite for further hit-to-lead and lead optimization stages of drug discovery. In such circumstances, starting a new drug discovery project with peptides close to the affinity and specificity of the innate substrates is a reasonable option. Rebollo et al. [14] found certain peptide macrocycles, sharing a resemblance with the sorting LPxTG sequence, possessing activity in low micromolar range against SrtA. Recently, Wang et al. [15] reported an oligopeptide LPRDA that is similar to the sorting sequence LPxTG and revealed the activity both in SrtA in vitro (IC50 = 10.61 µM) and in vivo experiments. A promising antivirulence profile of LPRDA with low general bacteria cell toxicity is confirmed by the high value of the minimum inhibitory concentration (MIC) of more than 200 μM, which should not cause evolutionary pressure. Importantly, the work of Wang et al. [15] includes a section devoted to molecular modeling of the LPRDA–SrtA complex.

Our long-term aim is to use our expertise gained in covalent [16] and non-covalent small molecule [9,17] SrtA inhibitor development to start a new project using LPRDA oligopeptide as a base and, first, reveal the key interaction patterns for the LPRDA–SrtA complex, and, second, by gradual modification, come to a more potent and more drug-like lead compound [18] using SBDD methods and subsequent experimental validation. In this work we make an effort to establish protocols pertinent to computer-aided SBDD for further rational modifications and development by starting from the molecular modeling section of the work by Wang et al. [15]. The reliable binding mode of LPRDA to SrtA of *S. aureus* is crucial to stemming further rational development both using docking—for initial placement of LPRDA analogs—and more resource demanding relative free energy methods, such as FEP and TI [19,20,21]—to accurately estimate the free energy changes.

In what follows, we first validate the binding mode quality at a structural level. To that end, we composed a list of criteria, based on a critical reflection on the diverse sources of currently available information. Next, we describe the docking efforts aimed at obtaining the reliable binding mode. The molecular dynamics studies were carried out in order to refine the geometries and estimate the stability of the complexes, predicted by docking. Finally, the hypothesis about SrtA binding is formulated and the future prospects and conclusions are derived.

## 2. Results

### 2.1. PDB Structure Validation

StrA has been shown to be a tough target due to the presence of the intrinsic large-scale conformational motions, with no apparent best mode revealed yet to bind the sorting signal LPxTG or its close analogs [6]. Our aim is to establish a reliable in silico protocol to employ SBDD, firstly to establish the key interactions, and secondly to iteratively optimize the LPRDA oligopeptide in order to come to a more drug-like substance, having both significant affinity and the corresponding ADMET profile. To that end, not only the PDB structure reported as being used for modeling by Wang et al. [15], but all the known experimental structures of StrA available in the PDB were analyzed to choose the most relevant one for further modeling. Currently, seven structures of *S. aureus* SrtA are present in the PDB (Table 1).

#### 2.1.1. Evidence from the Literature

##### The Intrinsic Motions and Partial Disorder

The uneven mobility of the different secondary structure elements of StrA was well reported in the literature, using both the experimental and simulation evidence [6]. Whereas the eight-stranded β-barrel fold undergoes the least thermal fluctuation, the β7/β8 and especially β6/β7 show significant movement to the degree that the β6/β7 loop was classified as an intrinsically disordered region (IDR) of SrtA, which undergoes a disorder-to-order transition upon ligand binding [7].

The presence of Ca^2+^ in its binding site within SrtA was shown to cause the restriction in the mobility of the loops and hence the intrinsic disorder [25].

Overall, the appreciable difference in the binding sites observed in PDB structures is mainly described by the different conformations of the β6/β7 and β7/β8 loops. The motion of those loops was shown to correlate significantly [6].

##### The Use in Molecular Modeling

Many of the PDB structures from Table 1 have been used in molecular modeling of SrtA internal dynamics and its interaction with different ligands, both via docking and molecular dynamics tools; the most used PDBs are 2KID, 1IJA, 2MLM, 1T2P and 1T2W. Thus, one can see that despite the binding sites in the above protein models differing significantly, each of the forms seems to have found its use in rationalizing ligand-receptor interactions. One may fairly argue that the approach is not strict. However, there is a sound justification behind this simplistic approach as well. Whereas the β6/β7 loop is partially disordered and the β7/β8 loop can adopt several conformations, many of their conformations are feasible and certain conformations are more amenable for binding particular ligands than the others. In case of non-covalent inhibitor development, the catalytic event is not necessary and is even not desirable, thus the specific arrangement for the catalytically “active” conformation should not be pursued. The latter broadens the energetically accessible conformations of SrtA which could provide good binding of a ligand.

#### 2.1.2. Visual Analysis

Historically visual ligand-receptor complex analysis performed by experts in the field has played, and still plays, a crucial role in drug discovery [26]. It both allows researchers to discard unrealistic binding modes and helps to obtain additional complex insights regarding which direction to better proceed with the discovery. In this work, we initially aimed at finding the structural motifs of experimental complexes, which could be used as additional validation criteria for the modes obtained by docking. However, a critical look at the structures deposited in the PDB, using the expertise gained in drug-like molecules interaction with different pharmaceutically relevant receptors, has led us to several observations which may be generally useful for the rational design of molecules active against SrtA.

##### The Influence of the β6/β7 Loop Conformation

It is known that the presence of Ca^2+^ results in a closed form of the β6/β7 loop [7,25,27] and hence a significantly more compact binding site is observed, e.g., PDB:2KID. We hypothesize that such a compact form binding site structure is more appropriate for the development of small molecule inhibitors, than for the initial search of the plausible oligopeptide-SrtA complexes by docking due to the excessive conformational restrictions of the flexible site. While it was established using a kinetics analysis [27] that Ca^2+^ promotes substrate binding, hence the closed form of the β6/β7 loop being crucial, the enforcement of the closed form of the site in docking modeling at early stages may lead to missing the feasible oligopeptide-SrtA modes due to non-exhaustiveness and inaccuracies of the sampling and optimization algorithm used in docking.

##### Polypeptide Containing Structures

Structures in PDBs 2KID and 1T2W both contain as ligands the sorting oligopetide sequences, closely resembling the sorting sequence LPxTG, and thus presumably may serve as a source of crucial insights into oligopeptide-SrtA interactions. The available experimental structures of *S. aureus* SrtA have been thoroughly analyzed and compared to both the sortases from different species as well as to the available non-structural experimental data [28]. In particular, Thr-in and Thr-out (T in LPxTG) binding conformations were discussed in detail, and it was pointed out that in the Thr-out conformation observed in PDB:2KID, the Thr of LPxTG points out into the solvent, whereas the methyl group of the side chain of Ala(A) of the Cbz–LPAT* ligand (in PDB:2KID) points into the protein surface, which heavily contradicts with the arbitrariness of the amino acid in this position (x in LPxTG). Another point of concern is that T* from Cbz-LPAT* contains an additional C–S bond compared to the tioacyl bound Thr (T) for the classical substrate. In the absence of an additional bond, the oligopeptide sequence should shift by ca 1.8 Å (a mean C–S bond length) closer to Cys184, with possible binding mode re-arrangement of the other residues.

The Leu and Pro residues of the Cbz–LPAT* ligand (PDB:2KID) adopt rather reasonable conformations within the binding site, with the Leu side chain tightly occupying a hydrophobic pocket, formed by the β8-strand and partially ordered β6/β7 loop. At the same time, the phenyl moiety of Cbz N-terminus of the oligopeptide points mostly into the solvent, which is not natural by itself, and moreover it implies the full length peptide chain (of the natural substrates), preceding L of LPxTG, also points into the solvent, which is hardly the case. The reasonable directed interactions of the Cbz–LPAT* ligand are also lacking in PDB:2KID. The hydroxyl group of the Thr side chain only establishes a hydrogen bond with His120. The carbonyl oxygens of Leu, Pro and Ala of the ligand point toward Arg197, with no prevailing single conformation with well-established hydrogen bonds between those oxygens and the guanidine moiety of Arg197. Still, in NMR-derived PDB:2KID in several conformations, Arg197 forms more geometrically conventional hydrogen bonds with the above carbonyl oxygens. Thus, this pattern can be considered as entropically favored electrostatics interaction, where multiple nearly equivalent options exist to establish several directed hydrogen bonds. Overall, due to the provided reasons, PDB:2KID structure cannot be considered as a source of a structural template to compare the binding conformation for a general LPxTG-like polypeptide. Nevertheless, several interaction patterns seem rather general and useful.

PDB:1T2W is the second source of experimental information which may potentially shed light on the binding of signal oligopeptides to SrtA. It is a catalytically inactive C184A mutant SrtA non-covalently complexed with LPETG oligopeptide. This complex could be considered as a very relevant source of structural insights, but a careful visual inspection reveals several significant misfits. First, one of the C-terminus carboxylate oxygens is located as far as 3.3 Å away from its carboxylate carbon neighbor. The relative geometry is also not preserved. This oxygen atom is most probably due to the water molecule, but in this case the absence of the carboxylate oxygen is crucial. The overall position of the presumably charged C-terminus carboxylate is questionable: there is no apparent positively charged, or at least hydrogen donor, moieties near it in the PDB structure, which would explain why it is not directed into the solute. In this regard, it is also interesting that positively charged Arg197 is coordinated by neither C-terminus nor the side chain Glu (E) of the LPETG oligopeptide, but rather with a single carbonyl oxygen of the main chain of Glu. Second, the negatively charged side chain carboxylate group of Glu is surprisingly located in the Ca^2+^ binding site, in the direct vicinity of the side chain carboxyl group of Glu105 (2.6 Å being the closest distance between a pair of oxygen atoms). Interestingly, no water molecules were identified in the PDB structure in between the two carboxylic groups, which thus make a direct contact, also seen by VDW radii intersection by visual inspection. The latter contact between the same charged groups is not usual, so it requires additional explanation and thus could hardly be considered as a reliable structural feature of the considered complex.

Aside from the already mentioned Thr-out conformation of the LPETG ligand, the spatial location of E residue, which could be any (x) in LPxTG, does not support the arbitrariness of the aminoacid in this position, since the substituent at Cα position is directed towards the protein surface. The placement in the PDB:1T2W complex of the hydrophobic and highly conserved side chain amino acids L and P of the LPETG oligopeptide can hardly explain their role in the experimental activity measurement. Whereas P residue establishes more or less explainable hydrophobic contact with a small pocket formed by the β6/β7 loop, the L hydrophobic side chain is completely located in the solute, which is simply unnatural (besides contradicting the activity experiments). This discrepancy was also noted previously [6]. Starting from the available LPETG position it is hard to reasonably continue the polypeptide chain from both N- and C-termini. In the former case, the chain is located in the solution, whereas in the latter case the C-terminus is deeply buried in a small pocket with mostly neutral electrostatic potential. We agree that the relation to the features important to catalytic activity can be argued, since not the catalytic activity but rather the ligand affinity is being sought in this case, but, at least, the structural characteristics should be readily interpretable in terms of intermolecular interactions used in medicinal chemistry [29] in order to establish further rational discovery.

##### Other Complexes

The remaining complexes (Table 1) are split into two groups: (1) the apo forms of SrtA (PDBs:1IJA, 1T2P, 1T2O) and (2) SrtA bound covalently to small molecules (PDBs: 2MLM, 6R1V). The relevance of those complexes to LPRDA–SrtA modeling is questionable due to the significant flexibility of the β6/β7 and β7/β8 loops (both main and side chains), which establishes the possibility of SrtA to adapt to the substrate/ligand bound. The visual inspection of many SrtA geometries deposited in the PDB (including a set of 20–25 distinct models in NMR complexes) confirms the above variability, which contrasts sharply with the rigidity of the β-barrel core of the enzyme. The observed mobility of the β6/β7 and β7/β8 loops in the experimental geometries is in full accordance with the previous molecular modeling results (see Section 2.1.1. “The intrinsic motions and partial disorder”). In particular, for apo form complexes, the β6/β7 loop tends to adopt a conformation, where the mainly hydrophobic pocket between this loop and the β8 strand is small or even not formed at all, thus ruling out a possibly favorable interaction with a ligand. The generally more compact forms of the binding site of apo form geometries can be considered as a self-interaction of the part of the β6/β7 loop with that pocket. Similarly, for the covalently bound small molecule complexes, the combination of dissimilar to LPRDA ligands with the enhanced flexibility of the mentioned loops results in geometries which are very likely to be “overfit” to the specific ligand, and less acceptable for modeling interactions with a general ligand.

Therefore, the choice of the SrtA geometries, which are specifically adapted either for certain small molecule ligands or their complete absence (apo forms), is not well warranted. From this perspective, the most reasonable choice is to use SrtA conformation, which is both the most “open” among the experimental ones and the most relevant to polypeptide ligand binding. Among the known (Table 1) experimental geometries the most relevant thus is the PDB:1T2W structure, which was chosen for the subsequent modeling.

#### 2.1.3. Ligand–SrtA Complex Structure Validation Criteria

The analysis of the available PDB structures does not provide strong arguments not to follow the modeling procedure described by Wang et al. [15] and stem the modeling from the other protein structure. Despite an assumption of the single conformation of SrtA protein being evidently a weak point, in view of the significant mobility of the β6/β7 and β7/β8 loops, we hope to, first, reproduce the modeling and, second, potentially find a representative set of conformations for subsequent rational drug discovery. As an additional result, the conducted visual inspection helped us to formulate the structural criteria, which can be applied to the analysis of polypeptide–SrtA complexes, obtained during the simulation via either molecular docking or molecular dynamics.

Whereas the SrtA binding site dynamics is relatively well studied [6,7,28], a brief survey of molecular modeling efforts for non-covalently bound small molecules reveals quite different binding modes and interaction patterns, with notably few persistent directed interactions, such as hydrogen and halogen bonds and salt bridges. Therefore, we composed a list of structure validation criteria to consider when assessing the feasibility of the ligand–SrtA complex, taking into account its complicated nature:The same and opposite charged groups proximity;Location of hydrophobic parts/residues—in pockets or in the solute;Reasonable placement of N- and C-termini, assuming continuation of the polypeptide chain;Reasonable proximity of the catalytically broken bond of an oligopeptide to Cys184 (or its place in case of C184A mutant)—the bonds between T and G in LPxTG and between D and A in LPRDA;If the position of binding close to the catalytically relevant one is sought, the side chain of x in LPxTG should not be restricted by the protein surface;If possible, Arg197 should participate in hydrogen bonding with a ligand as hydrogen bond donor [9,30].

This set of requirements is not strict or exhaustive, but at least it rests on both the experimental information available and “small molecule”–“protein” interaction experience and provides a common framework for comparing different binding modes of ligands.

### 2.2. Molecular Docking

#### 2.2.1. AutoDock Vina

The molecular docking conducted with the default AutoDock Vina settings resulted in a series of binding modes with a narrow range of estimated binding energies (from −6.6 to −7.3 kcal/mol for 15 different binding modes). A quick visual inspection reveals that the vast majority of the binding modes (13 out of 15), although differing, have much in common—LPRDA is located in the binding site in the reversed polypeptide sequence orientation as compared to the well-established sequence position for the sorting LPxTG sequence analogs. Only two modes found resemble the direct sequence order of LPRDA in the binding site of SrtA (Figure 1).

In most positions found by AutoDock Vina, the ligand’s Arg residue is located in the Ca^2+^ binding site, where it establishes a network of salt bridges and hydrogen bonds with side chains of Glu105 and Glu108, as well as with the carbonyl oxygens of the main and side chains of Asn114, and occasionally with the hydroxyl oxygen of the Ser116 side chain. In many of the positions (including those presented in Figure 1), the LPRDA ligand also establishes favorable geometry salt bridges and/or hydrogen bonds with Arg197, which was shown as important for activity and ligand binding [30]. In two detected “direct sequence” binding modes, Arg197 is coordinated by either side chain Asp (D) or C-terminus Ala (A) carboxylate group. In the “reverse sequence” binding modes, Arg197 is coordinated by either one of the same carboxylate groups (but from different directions) and/or by the main chain carbonyl oxygens of Pro (P), or occasionally Arg (R). Those interactions for two different sequence orientations are comparable in quality and number, and thus should be comparable in energy. The main difference, according to visual analysis, is due to how the hydrophobic pockets of the site are exploited. In the case of the “direct sequence” binding modes, the hydrophobic parts of L and P of the ligand tend to occupy the hydrophobic pocket formed by the β6/β7 loop and β8-strand. Whereas in the “reverse sequence” binding modes, the Leu (L) side chain tends (a very frequent pattern among different modes found) to occupy a small hydrophobic pocket, formed by the aliphatic part of Arg197 and C184A residue. Interestingly, this L side chain position resembles closely the “Thr-in conformation”, which is believed to be crucial for pre-catalytic events [28].

The discovered complexes mostly meet the requirements of the structure validation criteria proposed above. For instance, the charged groups are reasonably placed within the site, Arg197 is coordinated with the directed non-covalent bonds, and the hydrophobic parts of the LPRDA ligand are reasonably placed within the SrtA binding site. On the other hand, the positions found do not generally provide well explainable ways to continue the LPRDA polypeptide sequence from both ends. Furthermore, perhaps the most significant disagreement is in the prevailing “reverse sequence” binding mode, where the Asp (D in LPRDA) residue, which is presumably similar to Thr (T from LPxTG), is located far from the C184A catalytic residue, in contrast to the “direct sequence” binding modes.

To compare the results with the work of Wang et al., the quite close interaction energy values are obtained in our work. The entire range of 15 complex geometries found by docking (−6.6 to −7.3 kcal/mol) is close to the value of −6.9 kcal/mol reported by Wang. Moreover, in particular, the two “direct order” sequence binding modes found in our study have the affinity estimates of −6.9 (#7) and −6.8 (#11) kcal/mol, which perfectly coincides with Wang’s results.

Due to the significant contribution of the electrostatic interactions apparently involved in LPRDA–SrtA complex formation, and the notion that Wang et al. obtained the complex geometry to start MD simulation using AutoDock, we decided to use AutoDock as well to study the same complex formation. AutoDock directly evaluates the Coulomb electrostatic interactions via point charges, which could provide more relevant insights into the highly electrostatic-driven environment.

#### 2.2.2. AutoDock

The docking results obtained with AutoDock reveal the elevated importance of electrostatic interactions, as expected. Moreover, the importance of the electrostatic interactions seems to be exaggerated in this case. On the one hand, in almost all modes with favorable interaction energies, both Arg197 and Glu105 (the closest of the Ca^2+^ binding site residues) are multiply coordinated with the LPRDA ligand. On the other hand, in most of the modes found, the conformation of LPRDA is heavily affected by a dense network of intramolecular hydrogen bonds and salt bridges. Although the latter could be expected under in vacuo conditions, the water medium does not favor such numerous intramolecular electrostatic-driven interactions. However, there is also a grain of truth behind that behavior. When the highly polar groups of the LPRDA ligand are mutually coordinated, the external polarity of the ligand lowers significantly, making it more resemblant of small molecule drugs due to a reduction in the accessible polar surface area and the number of accessible hydrogen bond donors and acceptors. Another reason is that the more compact and stable form of the ligand results in a lesser entropy penalty upon binding to the SrtA receptor.

The estimates of the free energy of binding for the standard AutoDock protocol are lower in absolute values (Figure 2) than for AutoDock Vina, although comparable by the order of magnitude.

The binding patterns are very different, with no apparent predominance of the “direct order” sequence binding mode.

We decided to tweak the search settings in order to increase the exhaustiveness of the plausible binding mode search for two reasons. First, the classical AutoDock Lamarkian genetic algorithm [31] is now considered to be not very efficient in general. Second, the number of rotatable bonds in LPRDA, 17, significantly exceeds that number among the small molecule drug-like molecules, for which AutoDock was initially developed.

### 2.3. Protocol Refinement

The default protocols for both AutoDock and AutoDock Vina were refined to increase the exhaustiveness of the search because of the high number of rotatable degrees of freedom in the LPRDA ligand. Thus, a hypothesis is that the unexpected results obtained above could be attributed to insufficient quality of search. The exhaustiveness of the search was modified separately for AutoDock Vina and AutoDock as described in the corresponding Method section.

#### 2.3.1. AutoDock Vina (ADV)

The increase of the exhaustiveness parameter for ADV did not result in any significant changes in the obtained results. The free energy estimates for the 15 final modes lie in the range of −6.5 … −7.2) kcal/mol, similarly to the default settings results. Visual inspection of the binding modes revealed only one mode, which could be attributed to the “direct sequence” binding of LPRDA to the SrtA binding site. The other modes are definitely the “reverse sequence” ones. The overall positions looked reasonable. From 6 to 9 intermolecular and from 0 to 4 (LPRDA) intramolecular hydrogen bonds were found in the binding modes. The partially hydrophobic parts of the ligand tended to occupy the corresponding places in the binding site. A certain neglect of long-range electrostatic account was still observed for several binding modes found by ADV, where the same sign formally charged residues of LPRDA and the residues Arg197/Glu105 of the SrtA receptor were located close in space, which is generally unlikely.

A possible explanation of non-changing docking results upon increasing the exhaustiveness is that even in the initial docking parameters the exhaustiveness had an already increased value. In any case, the lack of exhaustiveness does not explain the observed results.

#### 2.3.2. AutoDock (AD)

The increase in the number of genetic algorithm runs as described in the corresponding Method section resulted in that the most favorable complexes found have free energy estimations in the range of −6.0 … −7.9) kcal/mol, which is substantially lower than in the default settings AD run, which resulted in the best complex structure energy of −5.6 kcal/mol. The most favorable complexes are less tightly bound by intramolecular bonds, while forming reasonable intermolecular hydrogen bonds and salt bridges interactions. Therefore, the increase of exhaustiveness was necessary to find both more energetically favorable and better-looking (at expert visual inspection) complexes. For certain complexes both Arg and N-terminus ammonium positively charged moieties coordinate to the Ca^2+^ binding site, establishing multiple salt bridge and hydrogen bonds interactions. Similarly, in certain complexes both carboxylate groups of C-terminus and Asp residue of the ligand established interactions with Arg197. Generally, the modes obtained using AD were more diverse than those obtained using ADV, but a good portion of those diverse conformations looked unnatural at visual inspection. For example, the hydrophobic parts of the LPRDA ligand were not regularly placed in the corresponding pockets on the binding site surface, although the salt bridges and hydrogen bonds with good geometry were always established, which is the consequence of the exaggeration of the role of electrostatic interactions. The same analysis for ADV conformations yielded much fewer suspicious conformations.

As for the case of the ADV results, changing the exhaustiveness settings did not change the main qualitative result—most complexes were bound in “reverse order” sequence order, with very few complexes resembling “direct order” sequence modes.

### 2.4. Molecular Dynamics (MD)

#### 2.4.1. Docking Poses

During the docking studies it was found that it is hard to select a single mode out of a few reasonable binding modes, which could reasonably explain the experimental activity of LPRDA revealed in Wang et al. study. Taking into account the known flexibility of the β6/β7 and β7/β8 loops forming the binding site shape, it was decided to conduct MD studies in order to refine the initial geometries obtained by docking. Two hypotheses were tested. First, if the PDB:1T2W SrtA receptor geometry is not well suited to explain LPRDA–SrtA binding and activity, then MD refinement can result in a more suitable receptor geometry that could be used in subsequent modeling studies to find more active and stable peptidomimetic analogs of LPRDA. Second, if molecular docking results are not sufficiently reliable for this challenging system, then MD refinement will be able to distinguish between the spurious and more feasible binding modes. The former should be less stable within the more detailed and realistic force field-based MD simulation as compared to fast and rough estimates of interactions by molecular docking.

Several complexes in both “direct order” and “reverse order” sequence modes were selected for the subsequent study (Figure 3). Five initial complex geometries (two with “direct order” and three with “reverse order”) were taken from the ADV run with default settings. Three complexes (two with “direct order” and one with an alternative, Ca^2+^ binding site chelating geometry) were taken from AD results with increased exhaustiveness of search.

Each of the eight complexes were prepared for MD study using the standard protocol, in which the initial pressure equilibration (NPT part) is completed using the positional restraint on the coordinates of all heavy atoms of the complex in order to adjust the positions and equilibrate the solvent water molecule positions. After that, three independent replicas for each complex were run for the production MD using the same starting coordinates and different random velocities, generated for T = 300 K using the internal GROMACS means. That independent replicas run approach appeared crucial to reveal the inherent complexity of the system in question (Appendix A).

It is interesting to note that the flexibility of the β6/β7 and β7/β8 loops, which was earlier established by other researchers for StrA dynamics at long simulation times (1–10 µs) [6,7,25], is well observed even at timescales of the current modeling (10 ns). The final geometries of all 8 × 3 runs (after 10 ns) aligned with the initial structure (Figure 4B) are presented in Figure 4A. It is well seen that the β7/β8 loop (red loop in Figure 4A) spans conformational space near the initial conformation of the “closed” form (in a classification proposed earlier [6]). One of the conformations of that loop is apparently in the “open” form. The β6/β7 loop (orange in Figure 4A), in accordance with its partially disordered nature [7], spans an even larger conformational space. A random coil structure is observed partially or entirely for this loop. At the same time one to two turns of either 3_10_- or α-helix are observed in different places of the loop.

The intrinsic mobility of the main chain of the β6/β7 and β7/β8 loops, as well as of their side chains, defines the flexibility and variability of the binding site. In Figure 5 the surfaces colored with the simple electrostatic potential (using PyMol) were obtained for the conformations of the SrtA after 10 ns of MD simulation, aligned with the initial structure PDB:1T2W. It is clearly seen that not only the shape, but also the electrostatic/hydrophobic pattern of different pockets varies significantly. The common electrostatic pattern, shared among all the structures, is the positive potential (blue in Figure 5) of Arg197 and the negative potential (red) of the Ca^2+^ binding site residues. Thus, the above features should define the long-range recognition of the ligand, as well as represent the interaction features that do not greatly depend on a specific conformation of the loops of SrtA.

A detailed analysis of all the geometries obtained is out of scope of this study, however it is evident (Figure 6) that the LPRDA ligand spans quite different binding modes and orientations within the varying conformations of the SrtA binding site. It should be noted that despite the simulation time for the specified system being not enough to appreciably span the conformational space in order to single out the most favorable structures, it is enough (10 ns) to break locally unfavorable binding modes. Therefore, the involved interactions between LPRDA and SrtA are feasible. Again, as in the analysis of the docking structures, the mainly electrostatic nature of LPRDA–SrtA interactions suggests that the binding modes possessing the equal number of equivalent interactions should be roughly equivalent in energy up to very subtle modifications. This reasoning seems to explain the vast variability of the rather stable binding modes of the studied complex. A brief statistical analysis showed that on average 6.4 ± 2.0 (with the median number being 6) intermolecular LPRDA–SrtA hydrogen bonds are present in all 24 MD trajectories. Another consequence is that seeking a single “best” binding mode, in case several quite different binding modes contribute nearly equally to the overall free energy of binding of LPRDA to SrtA, is at least a hard problem and perhaps even not warranted at all.

Another confirmation of the generally non-tight binding of LPRDA to SrtA is the moderate to low values of the estimated ligand efficiency (LE), which are provided in Figure 3 for the docking complexes selected for subsequent MD simulation. The LE values span the range of 0.15–0.20, whereas the value of 0.30 is widely used as the practical threshold value for good small molecules binders.

#### 2.4.2. SrtA Apo Form Dynamics

There are two main hypotheses regarding what causes the observed enhanced conformational sampling of the SrtA geometry in the relatively short 10 ns MD of LPRDA–SrtA complexes obtained by docking. The first is that the initial SrtA geometry is not optimal for the water solution medium, due to it being more tuned to the crystal form (PDB:1T2W). Additionally, several simulation protocol options might also cause such behavior. The second hypothesis is that the presence of the ligand in the binding site causes the protein to sample the receptor’s conformations more rapidly, presumably in order to better adapt to the ligand. This hypothesis is partially supported by the diversity of conformations (especially related to the β6/β7 loop and partially to the β7/β8 loop) observed in the experimental structures, as well as by the previous state-of-the-art MD studies of the intrinsic conformational behavior of StrA carried out by the others [6,7,25].

Intrigued by our results from short time-range enhanced sampling, we decided to rule out the first hypothesis. To that end, the MD study of the apo form of SrtA was conducted in exactly the same settings and using the same preparation protocol, as was used to model LPRDA–SrtA complexes. Similarly, three independent 10 ns replicas of MD were started from the same NPT pre-equilibrated apo StrA system.

In a striking contrast to the results of MD of SrtA bound with the LPRDA ligand, the apo form of StrA did not reveal such a pronounced conformational sampling, with all the three independent replicas resulting in much closer conformations after 10 ns of the dynamics (Figure 7, Appendix A) than for the LPRDA–SrtA complexes. Nevertheless, the structural differences in the β6/β7 and β7/β8 loops was noticeable even in this case. Thus, both the initial SrtA geometry from PDB:1T2W and the protocol specifics do not account for the enhanced conformational sampling of SrtA complexed with LPRDA observed earlier. Therefore, the second hypothesis, that the presence of LPRDA promotes the conformational sampling becomes more plausible.

#### 2.4.3. Continued Dynamics

Our initial goal was to establish the “true” binding mode of LPRDA in order to stem rational development from it. The results obtained in both docking and short range (10 ns) MD simulations appeared perplexing for that purpose. Therefore, we decided to conduct longer MD simulations starting from several complexes in the hope that such simulation would relax partially spurious binding modes to the more consistent and stable geometries. For that reason, six initial geometries were selected by visual inspection from the results of 8 × 3 MD runs for 10 ns. One of the reasons for the initial geometry choice was to study if “direct order” or “reverse order” sequence geometries are preferred by the binding site of SrtA. Thus, several representatives of each mode were selected (Figure 8). The MD simulations were made with the same production settings as above but for 100 ns.

The results of the longer-lasting MD simulations are as follows. First, in four out of six runs, the LPRDA ligand remained in the binding site of the SrtA (Figure 8, Appendix A). In one run the ligand dissociated and in another run the ligand partially dissociated, leaving only the “chelating” contact of the positive side chain of Arg and the N-terminus of Leu with the Ca^2+^ binding site carboxylate residues (Figure 8G). Second, despite the fact that in the four other runs the complexes sustained 100 ns of MD simulation, the positions of the LPRDA changed substantially relative to the initial positions. The positions remained quite different with no evident binding motive prevailing. Third, the major interaction points were related to either positively charged Arg197 interaction or the interaction with the negatively charged Ca^2+^ binding site. For example, the intermolecular hydrogen bonds statistics for LPRDA–SrtA complexes for the four runs remaining in the binding site were 5.1 ± 1.7 (with median number being 5). Fourth, the RMSF analysis completed for the SrtA of all the 100 ns runs confirmed the enhanced mobility of the loops, in particular the β6/β7 and β7/β8 loops, related to binding (Appendix A). Fifth, the charged side chains of SrtA substantially altered the binding site electrostatic profile via occasional approaching to or distancing from the binding site. Sixth, the LPRDA ligand retained the ability to form and sustain the intramolecular salt bridges and hydrogen bonds, which was first revealed during the docking stage, and later re-discovered during the short MD simulation runs. Seventh, in one such self-associated conformations of LPRDA, the latter occupied only a half of the whole binding site, thus leaving space for the second molecule binding (Figure 8C). Eighth, even considering the loose definition, no clear preference toward either “direct order” or “reverse order” sequence binding mode of LPRDA could be established based on the results obtained. Thus, the problem appears even more complex than it would seem considering only the results of docking using a single target geometry. Definitely, LPRDA can adopt many plausible binding modes, none of which are decisive. This corroborates with the previous findings where LPATG was shown to bind at different places of the binding site of SrtA [6]. Taking into account the difference of the charged states of LPxTG and LPRDA, one can assume that the latter would bind more promiscuously due to fewer hydrophobic parts and greater number of polar contacts.

## 3. Discussion

Even at MD times as short as 10 ns there exists a wealth of different LPRDA–SrtA conformations. They are hardly the modeling artifact, since, first, different conformations result from the same starting complex geometry, and second, the modeling time is enough to relax the most unfavorable interactions if they were present in the initial complex geometries found by docking. Moreover, the apo form simulation in exactly the same conditions did not reveal such a pronounced conformational sampling of the SrtA target. However, even for 10 ns MD modeling of the apo form of SrtA, the three different replicas, started from exactly the same pre-equilibrated position, resulted in different conformations of the β7/β8 and especially β6/β7 loops. This fact supports the classification of those loops as intrinsically disordered and thus prone to chaotic motions at room temperature. The longer MD simulations (100 ns), continued for the selected conformations, also failed to reveal the very specific and long-time stable binding modes. Our findings corroborate with the previous work by Kappel et al. [6] where the LPATG polypeptide also resulted in many different binding modes. The main difference between the studies is the polar nature of the polypeptide ligand. Whereas the LPATG ligand from [6] only contained two formally charged N- and C-termini, the LPRDA from our study additionally contained two side chain groups (Arg and Asp) charged at physiological conditions. We believe the more charged form possesses higher ability to form multiple intermolecular interactions, each of which is not decisive or even strong. Thus, any two geometrically different binding modes differ in the network of specific hydrogen bonds and salt bridges but tend to have the comparable number of interactions, leading to a shallow free energy profile of the LPRDA to SrtA binding. Another distinction from Kappel’s work is the absence of Ca^2+^ in our work, similar to the work of Wang et al. [15]. However, it is well known that the presence of Ca^2+^ in its specific binding site of SrtA results in more restricted conformational motions and the preference of the more “active-like” forms of the conformations of SrtA [7,25].

The presence of multiple feasible but quite different StrA conformations resulted in difficulties in obtaining the best single binding mode of the receptor for further modeling. We believe that the accuracy of the force fields, although important [32], does not play the decisive role in that case, but rather the very nature of the SrtA receptor. Another consequence is that a simple a priori choice of the relevant SrtA conformation for modeling should be taken with a grain of salt.

An interesting result of our modeling is that when SrtA was complexed with LPRDA, the enhanced conformations sampling of the significantly flexible β6/β7 and β7/β8 loops took place. The first manifestation of the effect can be seen in comparing the structural results of the LPRDA–SrtA complex dynamics (Figure 4A) and the MD results for the apo SrtA form (Figure 7). It should be noted that for all those cases the same MD simulation conditions and the same initial SrtA geometry were used. Another argument toward the enhanced sampling in the LPRDA–SrtA complex is that in work by Kappel et al. [6] both regular MD and the accelerated MD did not result in numerous spontaneous transitions between the different conformations of SrtA during the considerable modeling time (100 ns) for the similar LPATG-SrtA complex. Our hypothesis to explain the “plasticizer” effect of LPRDA is that LPRDA has two additional formal charges and several additional freely rotatable bonds in the side chains of R and D compared to the LPATG polypeptide. That leads to the more flexible conformational space for LPRDA. In the realm of intrinsically disordered proteins/regions (IDP/IDR) it is well established that the more polar sequences tend to be more disordered. Another assumption is necessary to explain the observed difference in the flexible β6/β7 and β7/β8 loops sampling in the apo and holo (LPRDA) forms. It seems that a synergistic effect takes place when the disordered/chaotic nature of the β6/β7 and β7/β8 loops of SrtA and the disordered/chaotic nature of LPRDA combine in a complex. The network of easily interchangeable interactions (such as hydrogen bonds) facilitates the conformational transitions of each partner by readily adopting to partially stabilize the transition states. Evidently, a specific and more elaborate work is necessary to theoretically study the synergistic effect described. However, we believe the initial hypothesis provided above is plausible and at least is a good starting point for further investigation.

In accordance with the best modeling studies conducted earlier [6,7,25,33], StrA was confirmed to be a “hard target” for SBDD discovery due to the dynamic properties of its binding site. The lack of clinical candidates for this highly appreciated target makes an additional confirmation for the above statement.

Nevertheless, currently not all options to practically model SrtA to guide rational inhibitor design have been exhausted. In particular, the armory of approaches and tools developed in such highly related fields as protein–protein interactions (PPI) [34] and intrinsically disordered proteins/regions (IDP/IDR) [32,35,36] should be used more for SrtA modeling. In any case, the complex structures under questions should be critically evaluated using several criteria. Of course, any additional structural input from the experimental studies is indispensable for this particular realm.

## 4. Materials and Methods

### 4.1. Molecular Docking

#### 4.1.1. Target Structure

Wang et al. claim that the PDB:1T2P structure was used for modeling in their work, however a closer examination reveals that PDB:1T2W was actually used. The visual analysis done using the PyMol generated molecular surfaces colored with a simple built-in electrostatic potential estimation routine suggests that an almost perfect match is observed for PDB:1T2W and the Figure 4A from the work of Wang et al. (Figure 9b,c, as opposed to the pair Figure 9a,b). The geometry from PDB:1T2W provides a more open binding site of SrtA, thus more amenable for docking. Therefore, in our study we also decided to use PDB:1T2W as the source of geometry both for docking and molecular dynamics modeling. We do not anticipate the C184A mutation would be crucial for elucidating the binding mode of LPRDA, since this residue is located at the very rigid β-barrel structure and LPRDA inhibition is assumed to be non-covalent, so the (pre)catalytic geometries should not be of key importance.

The SrtA target for docking was prepared using a regular protocol. All molecules beside protein were removed from the structure. Chain A was used for the simulation. MGLTools v1.5.6 [37] with the standard settings were used to obtain pdbqt files necessary for both AutoDock and AutoDock Vina modeling.

#### 4.1.2. LPRDA Ligand Preparation

LPRDA polypeptide contains two amino acids with the charged groups inside the chain, as well as the two non-capped termini, which are also most probably charged at physiological conditions. For that reason, it was decided to model LPRDA in a fully charged state. It should be noted that such a charged molecule is not typical for best examples of small molecule drugs (Figure 10, Table 2). Whereas in in vivo experiments Wang et al. [15] reported the use of a PEG-ilated at N- and amide capped at C-termini version of the LPRDA oligopeptide, we adhered to the uncapped LPRDA, since in vitro activity in the Wang et al. paper was reported for this uncapped variant.

The reasons for possible LPRDA optimization are twofold. First, LPRDA is a polypeptide, thus subject to fast biodegradation by both bacteria and human host. Second, there is strong departure of the physico-chemical properties of LPRDA (Table 2) from those characteristics for orally administered drugs. Those requirements though could be relaxed in case of the intended external use of a developed medicinal product. In the latter case, a strong departure from the rule of five (RoF) [39] and other rules [40,41] is not essential due to direct delivery of the substance to the surfaces where it should act. A brief analysis of the drug-like properties of LPRDA and its analogs (Table 2, Figure 10) suggests that the polypeptide is too polar compared to the drugs that used to be considered as reasonably bioavailable. This is especially true for the fully charged form, which is likely to dominate at physiological conditions. Therefore, the ways to decrease polarity while preserving activity is the main direction of further optimization of LPRDA.

The LPRDA ligand was protonated at pH = 7 using OpenBabel v3.0.0 [38] and then prepared for docking using the standard protocol by means of MGLTools v1.5.6 [37], resulting in a pdbqt file.

#### 4.1.3. AutoDock Vina

In Wang et al. AutoDock Vina was reported to be used to screen different oligopeptides resembling LPxTG, which finally resulted in the best oligopeptide—LPRDA. For that reason, we also decided to use AutoDock Vina to find plausible LPRDA–SrtA geometries for further analysis.

The default parameters for docking were applied, except the number of reported conformations (--num_modes option) was increased from the default 9 to 15, and the exhaustiveness of conformational search --exhaustiveness option) was increased from the default value 8 to 32. The main reason for those corrections was the large number of rotatable bonds (17) present in the LPRDA molecule.

AutoDock Vina version 1.1.2 [42] was used for simulations.

#### 4.1.4. AutoDock

The default grid generation and docking parameters were used for the docking runs. The number of Genetic Algorithm runs (ga_run) was set to 100.

AutoDock version 4.2.6 (both autodock4 and autogrid4) [37] was used for simulations.

#### 4.1.5. Refined Experiment

Since the initial efforts to find a binding mode close to that reported by Wang et al. were unsuccessful, we tried to increase the depth of conformational search, since the default docking parameters are applicable to more drug-like small molecules with far fewer rotatable degrees of freedom. For that reason, the exhaustiveness parameter (--exhaustiveness) was increased from the initial value of 32 to 64 for AutoDock Vina simulation. For AutoDock simulations the initial value of 100 Genetic Algorithm (GA) runs (ga_run) was increased to 500. It should be noted that such an increase adds up a substantial computational overhead, resulting in an increase of computational time from 2 h up to 11 h per run. The other settings remained intact to facilitate comparison of the experiment results.

### 4.2. Molecular Dynamics

The selected LPRDA–SrtA complexes found by molecular docking were subjected to molecular dynamics (MD) simulation in order, first, to establish their stability, and second, to possibly find new, perhaps, more realistic binding modes, not accessible to docking.

All MD simulations were conducted using GROMACS 2022.1 version [43,44,45].

#### 4.2.1. Complex Preparation

Several conformations obtained in the docking studies were used to study by means of MD. To that end, the missing non-polar hydrogens (in the PDBQT representation of the ligand used in the AutoDock family of software) were added using OpenBabel v3.0.0 [38] software without modifying heavy atoms and polar hydrogen positions. While the protein force field could be applied to model the LPRDA polypeptide, it was intentionally decided to model it using the GAFF force field in order to lay foundations for further arbitrary modifications of the polypeptide, which would in any case eventually lose its initial peptide nature and protein force field types.

AmberTools22 [46] was used to get GAFF parameters for the ligand. Initially the AM1-BCC charge scheme [47,48] was tested, as it is the default charge scheme for use in GAFF. Unfortunately, the built-in AM1 optimization in this scheme led to conformations with multiple strong intramolecular hydrogen bonds and at least one salt bridge, resulting in quite perturbed partial charges for such interacting groups. The latter in particular leads to the non-integral charge of the formally charged groups due to electron density redistribution in salt-bridged groups. In order to avoid this artifact, it was decided to use MMFF94 charges [49] for further simulations. On the one hand, they were designed to reproduce the same level QC (RHF/6-31G*) molecular electrostatic potential, as was used in GAFF parameterization [50], sufficiently well. On the other hand, MMFF94 charges strictly preserve formal charges and ensure the topological symmetry of the charge values, whereas they do not depend on a conformation.

Ligand geometry and topology obtained by AmberTools was converted to GROMACS format using the ParmEd library [51] shipped within AmberTools. All subsequent processing was done using GROMACS utilities.

Protein structure taken from PDB:1T2W was converted to GROMACS format using the pdb2gmx utility. The AMBER14SB (amber14sb_OL15_cufix_zn.ff) force field was applied in the GROMACS environment used for the SrtA protein, with parameters taken from https://github.com/intbio/gromacs_ff (accessed on 8 August 2022). The TIP3P water model was used as a solvent. For different complexes 7744 to 7754, water molecules were added using gmx solvate utility with the modified value of VDW radii scale parameter (see below).

The entire LPRDA–SrtA system has the total charge +1 and was thus neutralized by a single Cl^−^ using gmx genion utility.

A slightly modified version of the standard protocol was used to prepare initial complexes. During the initial runs it was discovered that using the standard preparation protocol starting from the molecular docking geometries often resulted in a few water molecules being placed between the ligand and the surface of the protein site. Subsequent heating during the MD led to movement of those water molecules from the trapped hydrophobic states into the solute at the cost of displacing a part of ligand from the protein surface. To address this issue, the value of scaling VDW radii used by GROMACS to decide whether to place a solute molecule (-scale option to gmx solvate) was increased from the standard 0.57 to 0.62. It helped to remove the questionable water molecules in the space between the ligand and protein in the initial geometries. As a side effect, that also resulted in a depleted first shell of water molecules around the entire surface of the protein, so in order to compensate for the depleted number of water molecules, the box size was increased from 10 (default) to 14 Å. Octahedral periodic conditions were used for the MD simulation. In order to relax the influence of the procedure modification made, the time to model the NPT ensemble was increased from 100 to 200 ps, whereas the heavy atom positional restraints were relaxed from 1000 to 500 kJ·mol^−1^·nm^−2^ (2.39 to 1.20 kcal·mol^−1^·Å^−2^) values. The subsequent analysis confirmed that the protocol adjustments listed helped to avoid the disruptive effect of the trapped water molecules disputably placed into the initial complex geometries. On the other hand, as the NPT MD equilibrated geometry analysis of several complexes revealed, where water molecules are crucial, they had paved their way to the protein surface–ligand voids during the simulation, starting from the waterless void geometries.

The final complex preparation for MD was as follows. First, steepest descent ligand-receptor optimization was performed using the periodic boundary conditions (PBC) for each initial complex geometry with 50,000 steps allotted and the maximum force (emtol parameter) of 1000 kJ·mol^−1^·nm^−1^ (23.9 kcal·mol^−1^·Å^−1^). Next, NVT heating to 300 K was performed during 100 ps, using a 2 fs time step in conjunction with LINCS algorithm [52] applied to H-bonds, PBC, and modified Berendsen thermostat (V-rescale) with tau_t = 0.1 ps. After that, NPT density equilibration followed for 200 ps with 2 fs LINCS time step, PBC, thermostat setting as for NVT, and isotropic C-rescale barostat with tau_p = 2 ps and reference pressure (ref_p) of 1 bar. Thus prepared, preliminary relaxed complexes were further subjected to three independent production runs.

#### 4.2.2. Different Runs Analysis

Each of the partially equilibrated LPRDA–SrtA complexes was subjected to three independent MD production runs using randomly generated particle velocities corresponding to 300K for each run. Each run was 10 ns in length, using 2 fs LINCS time step, PBC, V-rescale thermostat as in NPT equilibration above and isotropic Parrinello-Rahman barostat with tau_p = 2 ps. As the subsequent analysis reveals, in many cases the independent runs resulted in appreciably different complex geometry evolution during MD and, hence, the final geometries. The same settings were used for the continued MD runs for additional 90 ns.

#### 4.2.3. RMSD and RMSF Analysis

The root mean square deviation (RMSD) and root mean square fluctuation (RMSF) analyses were performed using GROMACS capabilities over the obtained trajectories. In order to trace both the protein (SrtA) and the ligand (LPRDA) movement relative to the initial docking positions, the complex geometries were aligned using Cα atoms of SrtA as a reference. RMSF analysis was done for the continued 90 ns runs, separately for each of the 6 ligand-receptor complexes, taking into consideration only the fluctuations of the Cα atoms of SrtA.

## 5. Conclusions

The main goal of the work was to establish the protocol to robustly model LPRDA (and its subsequent analogs) interaction with SrtA of *S. aureus* based on the work of Wang et al. [15]. Despite the establishment of the simulation protocol appearing cumbersome, the useful insights about the nature of the interactions in a hypothetical LPRDA–SrtA complex were obtained. We believe these findings will shed some light in the field of the development of antivirulence drugs against *S. aureus*. First, it turned out impossible to reproduce the simulation procedure described in the work by Wang et al., which appeared to be lacking important simulation details upon closer examination. Second, the docking experiments—using both AutoDock and AutoDock Vina—consistently predicted that the most favorable binding modes of LPRDA in the binding site of SrtA of *S. aureus* adopt the “reverse direction” sequence orientation relative to the established order of the sorting sequence LPxTG in the enzyme. However, the energetic preference is not significant according to the estimations of the AutoDock and AutoDock Vina scoring functions. The subsequent molecular dynamics studies confirmed that such modes are not likely to be spurious. Therefore, there is a probability that the difference between LPxTG in the peptide sequence (substituted at both N- and C-termini) and the free form of LPRDA may result in different binding modes, still leading to the experimentally observed SrtA inhibition in both cases. Third, our molecular dynamics studies confirmed that the β7/β8 and especially β6/β7 loops (the latter is classified as being intrinsically disordered in the literature) were responsible for significant changes in the shape and polarity of certain patches of the binding site of SrtA. Therefore, quite different interaction patterns with a ligand can be observed. Fourth, the appreciable changes in the conformations of the β6/β7 and β7/β8 loops were observed even for three different 10 ns MD runs starting from exactly the same geometry of the apo form of SrtA. It confirms the significant chaotic/disordered nature of those loops, in accord to the literature. Fifth, it was surprising to observe that SrtA complexed with LPRDA seemed to sample the conformations of the β6/β7 and β7/β8 loops appreciably faster than it has been previously reported for the long-time modeling of the SrtA in both apo and holo (LPATG) forms. Thus, LPRDA is shown to act as a “plasticizer” for those loops. This effect deserves further theoretical investigation. However, our initial hypothesis to describe this behavior is that both counterparts—the (partially) disordered SrtA loops and LPRDA itself—reveal the chaotic movements due to the wealth of plausible and energetically close conformations in each of them. It seems their combination upon complex formation results in a synergistic effect, enabling both partners to faster sample the LPRDA–SrtA complex conformations. Sixth, both molecular docking and molecular dynamics results suggest that the free energy surface (as well as the potential energy surface) of LPRDA–SrtA binding looks like a shallow valley with numerous minima, nearly equivalent in energy with quite different geometries. The main origin is in the predominantly electrostatic nature of interactions, with little impact of shape complementarity and hydrophobic effect, which could better distinguish different conformations. In each MD snapshot analyzed, LPRDA forms ca 5 to 10 hydrogen bonds (part of which are in salt bridges) with SrtA and many more coming from the water medium. Thus, provided each hydrogen bond is roughly comparable in energy, the same number of hydrogen bonds can be established by quite a great number of alternative conformations. Finally, LPRDA was shown to intrinsically have several structurally different options to coordinate the two most persistent interaction sites: the positively charged Arg197 and the negatively charged Ca^2+^ binding site. Due to the similar set of interactions established in each binding position affecting those two sites, an expected energetic difference between those positions is only marginal. Therefore, it is hard to distinguish them unambiguously. Overall, our results show that the simple assumption that LPRDA binds similarly to the sorting sequence LPxTG is unlikely. Moreover, the search for a single “correct” LPRDA–SrtA complex geometry seems to be an ill-defined problem when using fast and approximate means of modeling. Thus, at the practical level, a more involved protocol to study structure-affinity relationships for the LPRDA analogs should be designed that takes into account the achievements in the field of protein–protein interactions (PPI) and intrinsically disordered proteins/regions (IDP/IDR), as well as enumeration of different plausible conformations of SrtA.

## Figures and Tables

**Figure 1 molecules-27-08182-f001:**
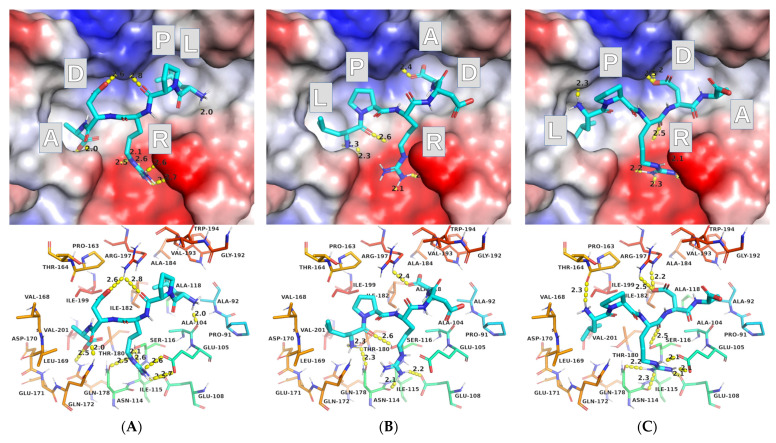
AutoDock Vina binding modes for LPRDA–SrtA complex: (**A**) mode #1—“reverse order”—“ADRPL” (from left to ring in the view), (**B**) mode #7 “direct order”—“LPRDA”, (**C**) mode #11 “direct order”—“LPRDA”. All images were made using the same view in PyMol.

**Figure 2 molecules-27-08182-f002:**
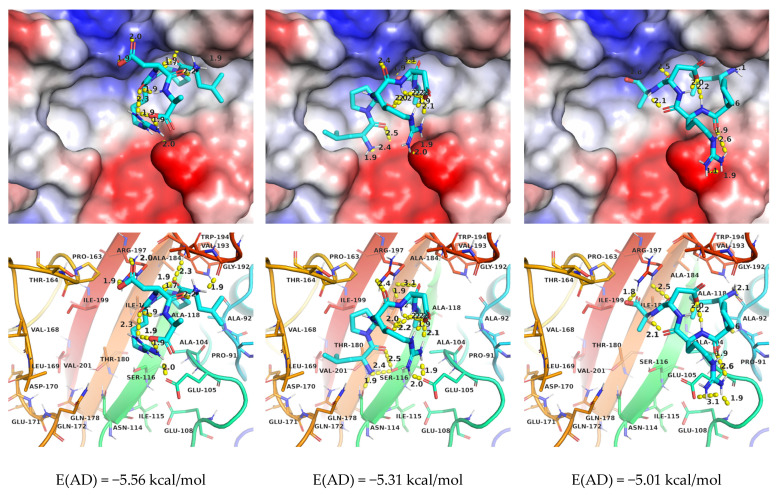
The first three binding modes of LPRDA-StrA found by AutoDock using the default search parameters.

**Figure 3 molecules-27-08182-f003:**
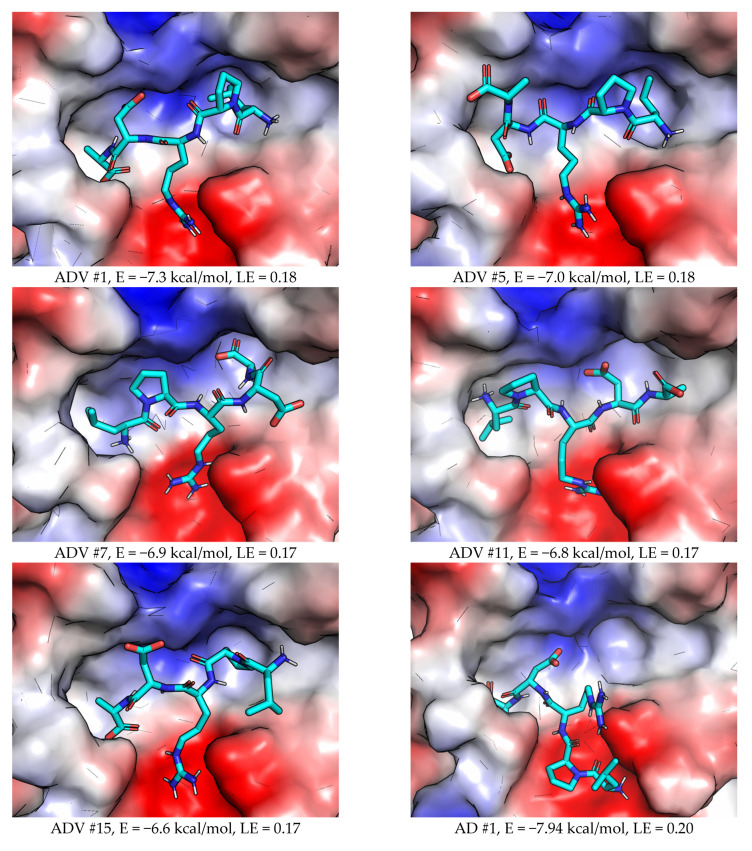
The initial LPRDA–SrtA complex geometries used for MD study. The scores (E) for either ADV or AD are provided for the best energy complex. The ligand efficiency metrics (LE) is calculated as LE = −E/NH, where NH—the number of heavy atoms of the ligand (for LPRDA NH = 40).

**Figure 4 molecules-27-08182-f004:**
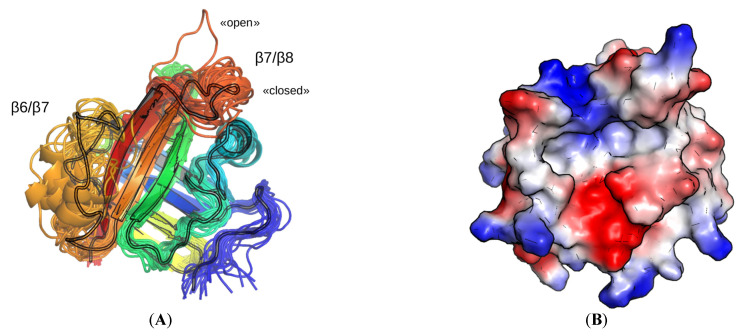
(**A**) Cartoon representation of the geometries of SrtA for each of the eight complexes and three replicas, overlayed onto the PDB:1T2W structure (shown outlined) using the function cealign of PyMol. (**B**) The simple electrostatic potential surface, generated for the PDB:1T2W structure in the same view as in the left panel.

**Figure 5 molecules-27-08182-f005:**
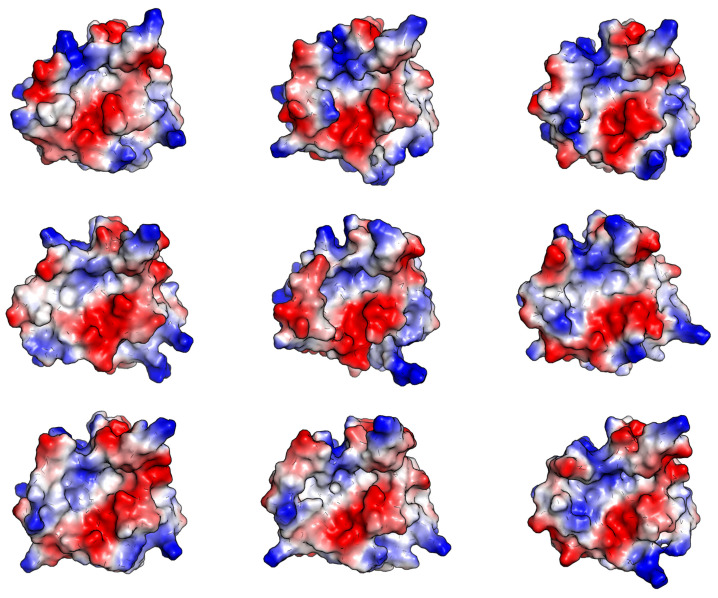
Different shape and electrostatic patterns of the effective binding site surfaces of SrtA obtained from MD simulation (eight complexes × three replicas each). The same alignment to the initial structure PDB:1T2W as in Figure 4 is used. Negative electrostatic potential is in red, positive potential is in blue.

**Figure 6 molecules-27-08182-f006:**
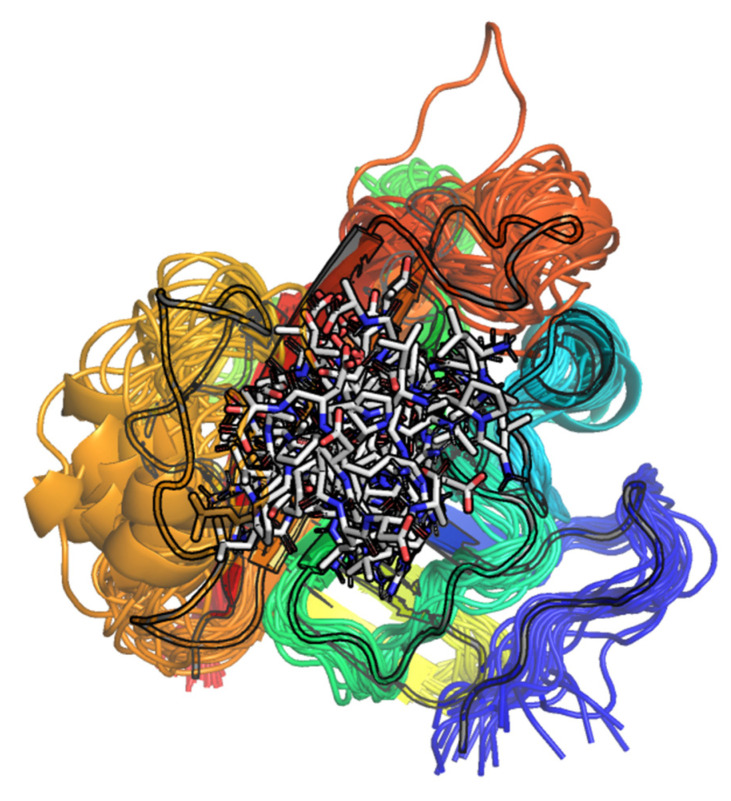
The resulting binding modes are different and occupy generally the entire binding site surface of SrtA. The same view as in Figure 4 is used.

**Figure 7 molecules-27-08182-f007:**
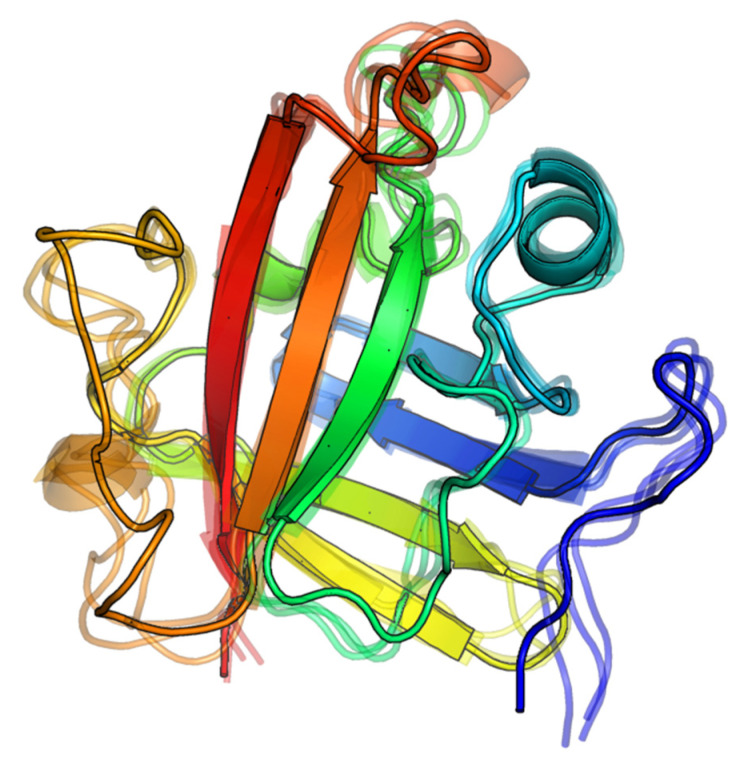
The initial and three final geometries of independent replicas of 10 ns MD of the apo form of SrtA.

**Figure 8 molecules-27-08182-f008:**
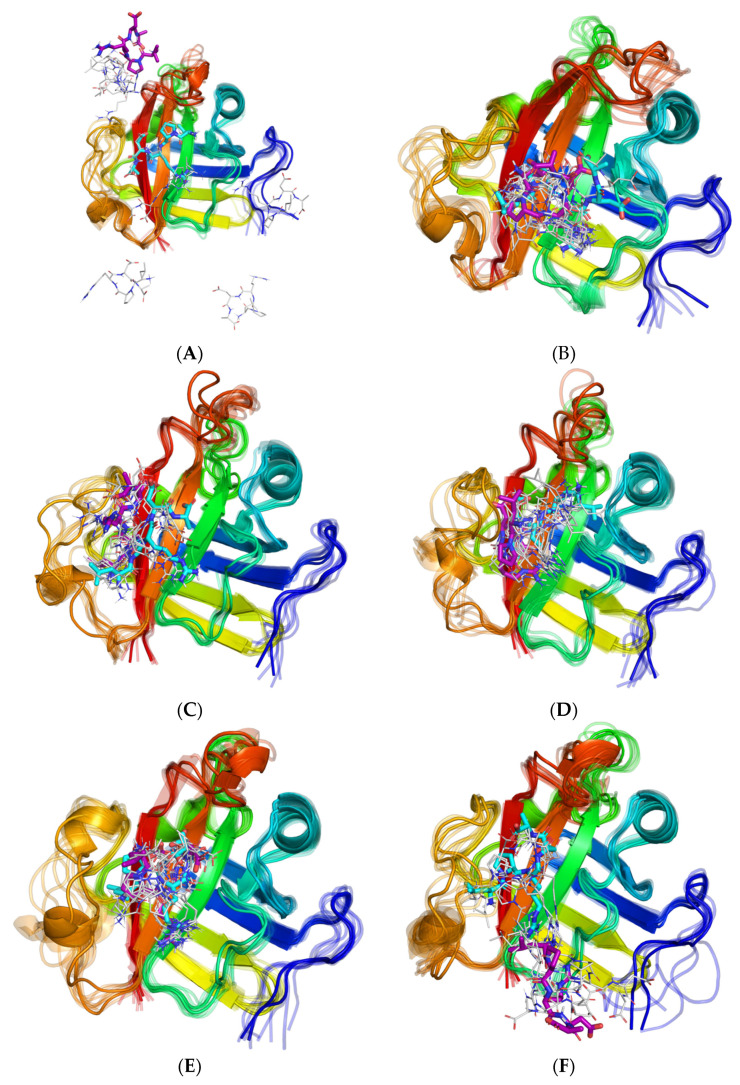
Initial (cyan), intermediate (gray, each 10 ns) and final (magenta) geometries of the “long” 100 ns dynamics for the six different geometries of the LPRDA–SrtA complex. The MD continued for: (**A**) ADV #1 (repl. 1), (**B**) ADV #7 (repl. 1), (**C**) ADV #7 (repl. 3), (**D**) ADV #15 (repl. 1), (**E**) AD #18 (repl. 3), (**F**) AD #20 (repl. 3).

**Figure 9 molecules-27-08182-f009:**
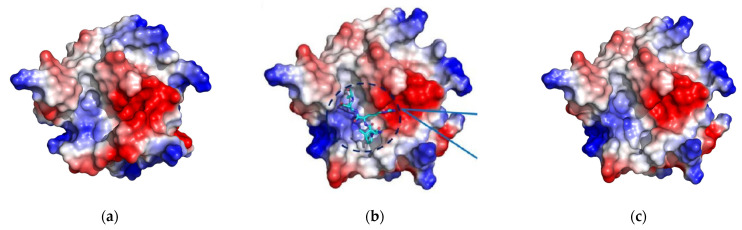
Surface models of PDB:1T2P (**a**) and PDB:1T2W (**c**) in comparison with the figure (Figure 4A) provided in Wang et al. paper [15] (**b**). PyMol was used to generate the images.

**Figure 10 molecules-27-08182-f010:**
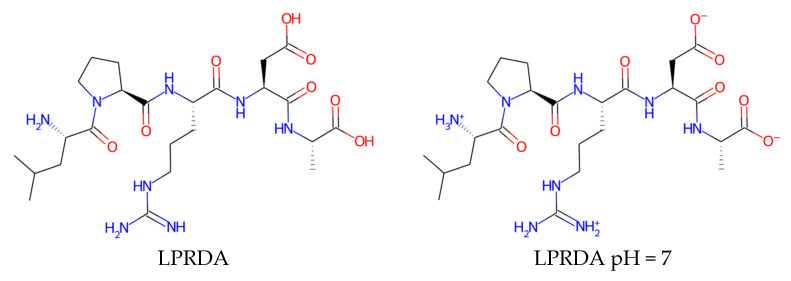
LPRDA in a fully charged state—both side chains as well as N- and C-termini.

**Table 1 molecules-27-08182-t001:** The experimentally determined *S. aureus* SrtA geometries from the PDB.

PDB	Source (Number of Modes)	Ligand	Ca^2+^	Bonding	Comment
2KID [22]	NMR (20)	Cbz-LPAT*	+	Covalent	-
6R1V [13]	NMR (20)	JPT	-	Covalent	-
1IJA [8]	NMR (25)	-	-	-	-
2MLM [23]	NMR (20)	2W7	-	Covalent	Strong variation of loop coordinates.
1T2P [24]	X-ray (1)	-	-	-	Three differing subunits. Crystallographic water.
1T2O [24]	X-ray (1)	-	-	-	C184A mutant. Three differing subunits. Crystallographic water. Two Met residues away from the binding site are replaced with Se-analogs.
1T2W [24]	X-ray (1)	LPETG	-	Non-covalent	C184A mutant. Three differing subunits. Crystallographic water. OXT atom (C-end) of the ligand has an unrealistic position.

T* is a threonine derivative that replaces the carbonyl group with -CH_2_-SH.

**Table 2 molecules-27-08182-t002:** Druglike properties of neutral/charged (at pH = 7) LPRDA and its N- and C-termini amide substituted analogs.

Name	Form	cLogP, OB *	cLogP, RDKit	HBD/HBA ^1^	TPSA	MW	#Rotors, OB ^1^
LPRDA	neutral	0.65	−2.35	9/16	270.13	570.6	21
LPRDA	pH = 7	−5.36	−7.56	7/14	279.15	570.6	21
(Ac)LPRDA(NMet)	neutral	0.57	−2.51	9/17	265.01	625.7	24
(Ac)LPRDA(NMet)	pH = 7	−2.68	−5.67	8/16	269.58	625.7	24

^1^ * HBA2 from OB; OB—OpenBabel v3.0.0 [38].

## Data Availability

Data is contained within the article or Appendix A.

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
