# Peer review of "Theoretical Studies of Leu-Pro-Arg-Asp-Ala Pentapeptide (LPRDA) Binding to Sortase A of Staphylococcus aureus"

_molecules, 2022, doi:10.3390/molecules27238182_

Round 1

Reviewer 1 Report

The authors looked into the PDB structures of Sortase A of Staphylococcus aureus. They docked the LPRDA peptide into one of the PDB structures using AutoDock, and selected several poses for further MD simulations. Results from the docking and MD simulations were analyzed in detail. Though their goal was to establish a protocol to model the LPRDA-SrtA complex, they concluded that the establishment of such a protocol is cumbersome, and the search for a single “correct” LPRDA-SrtA complex geometry is an ill-defined problem.

I do not recommend the publication of this manuscript, unless major revisions can be made, for the following reasons:

First, it is not realistic to me that the authors hope to discover a new drug molecule by gradually modifying a peptide. The authors should give examples of other drug molecules discovered in this way to convince the readers that this method is promising, or at least doable. On Page 43, the first paragraph describes that the physico-chemical properties of LPRDA are quite different from a drug molecule, which does not suggest this peptide is a good starting point. Nonetheless, studies of peptide-receptor interactions can be valuable for other purposes, e.g., protein degradation. But it’ll be best to use a force field suitable for peptides other than small molecules. After all, it’s the peptide-protein complex that the simulations were performed on.

As the author mentioned in this manuscript, AutoDock was initially developed for small drug-like molecules, not the best for peptide-protein docking. The authors may consider other docking tools, like ClusPro https://cluspro.org/login.php. The authors should test their settings to check whether docking LPETG into the apo would result in a pose similar to 1T2W, rather than just trying out default settings and exhaustiveness settings. Generally, docking software can provide many poses including the correct one (compared with crystal structures), but the ranking is not reliable. The free energies reported in this software are not accurate. Reporting these approximate free energies can be misleading. The authors selected a few docking poses for MD simulations. How can the authors be sure that they include the correct pose (compared to the crystal structure, if eventually there is one)?

MD simulations of 10 ns are too short to observe peptide-protein dissociation. If LPRDA does not bind SrtA well, the authors will not be able to see the peptide separate from the protein within 10 ns simulations. Since the authors have performed simulations of 100 ns, they should focus on results from 100 ns. In 2 out of 6 runs, the peptide dissociates from the protein. Does this suggest the binding is not stable? It seems that the authors avoided discussing the stability of the binding, but just simply said “LPRDA can adopt many plausible binding modes, none of which being decisive”. As there are a lot of conformational changes, how can the authors be sure that they have sampled the conformational space enough? The structures at the end of 10 ns do not provide stable binding modes. The authors may consider running the simulations longer, at least for several hundred ns, and using Markov State Models to analyze the structures from microstates.

Author Response

Q#1: First, it is not realistic to me that the authors hope to discover a new drug molecule by gradually modifying a peptide. The authors should give examples of other drug molecules discovered in this way to convince the readers that this method is promising, or at least doable. On Page 43, the first paragraph describes that the physico-chemical properties of LPRDA are quite different from a drug molecule, which does not suggest this peptide is a good starting point. Nonetheless, studies of peptide-receptor interactions can be valuable for other purposes, e.g., protein degradation.

R#1: We agree that the “small molecule way” has many benefits compared to the peptidomimetic one, but the problem is the SrtA target appears to be too flexible and having a shallow binding site in order to develop a meaningful small molecule lead compound. The small molecule lead compounds have not been reported for several decades, despite the target being considered prospective and relevant all the time. We have also spent considerable time designing small molecules covalent and non-covalent inhibitors of SrtA. In this work we decided to stem our development from the LPRDA polypeptide, which is reflected in the introduction section. Although the peptidomimetic route is generally harder, it is feasible. There are many examples of the leads and marketed drugs developed in such a way. For example, an excellent paper demonstrates the current status [10.1038/s41392-022-00904-4].

Q#2: But it’ll be best to use a force field suitable for peptides other than small molecules. After all, it’s the peptide-protein complex that the simulations were performed on.

R#2: It would be more consistent to use the protein force field for LPRDA, if we were not planning to design the peptidomimetics with generally arbitrary organic substituents. For that reason, we used GAFF from the onset. GAFF is a rather mature force field, and rough mistakes in its parameters are unlikely. Moreover, many parameters are shared between GAFF and AMBER (for proteins). The specific system LPRDA/SrtA is highly dominated with charged/polar interactions, which suggests the significance of the other parameters should be less pronounced. The intermediate conformations during the MD look very reasonable from the viewpoint of both intramolecular and intermolecular interactions (including their geometry) established. Thus we believe it is fairly valid  and wise to use GAFF for the study. We think that it's unlikely that the use of any other adequate alternative of the force field would result in significantly different qualitative results, compared to those we report.

Q#3: As the author mentioned in this manuscript, AutoDock was initially developed for small drug-like molecules, not the best for peptide-protein docking. The authors may consider other docking tools, like ClusPro https://cluspro.org/login.php.

R#3: Yes, we agree that it is a reasonable way to go. But, first, we were planning to reproduce the already reported modeling protocol by Wang et al. And, second, the main difficulty of LPRDA/SrtA system is not that the relevant complex geometry is hard to find, but rather that there is a wide continuum of binding geometries which are close in energy. Therefore, we speculate that most of the conformations exist at the same time with little preference for the specific ones. This finding corraborate with the results of the work by Kappel et at [10.1002/pro.2168]. However, they tried to express them in a more regular fashion, as if the specific complex geometries were found, not just rather vague clusters (as it was the case in their paper). So we believe that the probing with two widely used docking protocols with different search algorithms and scoring functions actually helps to reveal the real problem of the system under question.

Q#4: The authors should test their settings to check whether docking LPETG into the apo would result in a pose similar to 1T2W, rather than just trying out default settings and exhaustiveness settings.

R#4: Unfortunately, a meaningful validation using the innate ligand from PDB:1T2W is not feasible for the reasons discussed in the manuscript. First, the position from PDB:1T2W is hardly reasonable for the analysis, which is reflected in the section devoted to PDB complexes analysis. Second, Kappel et al [10.1002/pro.2168] arrived previously at the same conclusion, while analyzing this complex. Thus, since the complex geometry is itself under question, its use for docking validation is not warranted.

Q#5: Generally, docking software can provide many poses including the correct one (compared with crystal structures), but the ranking is not reliable. The free energies reported in this software are not accurate. Reporting these approximate free energies can be misleading. The authors selected a few docking poses for MD simulations. How can the authors be sure that they include the correct pose (compared to the crystal structure, if eventually there is one)?

R#5: Yes, we agree to the point in general but in this particular case the work done by us and the analysis of the most reliable results obtained by others for polipeptide/SrtA system allows us to state the following. For LPRDA/SrtA system the main problem is not how to find a single correct mode but rather the existence of seemingly shallow PES surface with multiple, nearly equivalent in energy and different in geometry, binding options. We believe that it's not the fallacy of a particular method/software, but rather is the property of the system under question. We believe it is tightly linked to the function of the SrtA, which is able to adapt to the signal sequence, then produce the catalytic event, then coordinate the second substrate and produce the final cross-linking. Another, more practical point is that LPRDA is also not quite a usual molecule (not a customary drug-like one), so its interactions may be not traditional for the field, which requires careful investigation rather than straightforward application of familiar tools. In accordance with the hypothesis above, we find a very shallow (estimated free) energy landscape for LPRDA/SrtA for two quite different SFs and search algorithms used in the software. The same basically applies to our MD study, where LPRDA was able to sample quite distant positions on the surface of SrtA. The lack of distinct positional preference of the LPRDA also agrees to the above hypothesis. So our answer is, even if the crystal structure appears eventually, it will most probably contain several positions of LPRDA (vaguely defined in electron density). But the useful findings from our analysis is the interaction patterns that most probably will be observed. They include salt bridge and hydrogen bond interactions with Arg197 and also multiple salt bridges and hydrogen bonds with several acidic aminoacids forming Ca2+ binding site. That is reflected in the manuscript.

Q#6: MD simulations of 10 ns are too short to observe peptide-protein dissociation. If LPRDA does not bind SrtA well, the authors will not be able to see the peptide separate from the protein within 10 ns simulations. Since the authors have performed simulations of 100 ns, they should focus on results from 100 ns. In 2 out of 6 runs, the peptide dissociates from the protein. Does this suggest the binding is not stable? It seems that the authors avoided discussing the stability of the binding, but just simply said “LPRDA can adopt many plausible binding modes, none of which being decisive”. As there are a lot of conformational changes, how can the authors be sure that they have sampled the conformational space enough? The structures at the end of 10 ns do not provide stable binding modes. The authors may consider running the simulations longer, at least for several hundred ns, and using Markov State Models to analyze the structures from microstates.

R#6: As already mentioned in the manuscript, 10 ns MD time is not sufficient to definitely check the stability of the complexes, but it's sufficient to uncover unfavorable intermolecular interactions, which could have resulted from the docking. Moreover, we surprisingly discovered that the same 10 ns of MD in conjunctions with LPRDA complexed resulted in the enhanced sampling of the b6/b7 and b7/b8 loops, which usually take far longer. Thus the system was able to get rid of the most unfavorable interactions and initially fit (via b6/b7 and b7/b8 loops movement) to the better modes. Several modes were continued just for 100 ns to ensure the initial findings are correct. It was the case. One of the 6 100 ns runs ends up with a reasonable (in terms of the interactions formed) position of LPRDA close to the binding site. We do not avoid discussing stability. We were planning to do so as a part of regular analysis in the field (which we've done a lot previously). The problem at technical level is the absence of the distinct positions, whose stability to analyze. Of course, we could select "some positions" out of the multiple MD trajectory and collect "some statistics" for them. But we believe it's unfair for the readers. So our main finding is there exists a shallow PES with multiple different geometries of LPRDA/SrtA. It's a fair and wise answer, resulting from our modeling efforts as well as the analysis of the literature for the close system modeling. On the basis of the above finding/hypothesis, since finding a "single and correct" binding mode is at least cumbersome and perhaps even not feasible, all other efforts toward this direction are out of the scope of the work, which was aiming at reproduction of the simulations by Wang and supplementing them by reporting the structural characteristics (as usual) and statistics (RMSD, MM/GBSA, per residue decomposition, etc.). Thanks for the advice, we think the subsequent work (we still need to establish a reasonable framework for interaction of LPRDA analogs with SrtA) will be planned with the more state-of-the-art approaches including Markov State Models, replica exchange MD and accelerated dynamics. We believe that the current findings of the work are useful for the scientific community. To better clarify our findings and vision we have reworked both Discussion and Conclusion sections.

Reviewer 2 Report

Shulga  and  Kudryavtsev,  examine  (LPRDA) to binding  SrtA ,  which conduct theoretical studies of the binding/activity of Leu-Pro-Arg-Asp-Ala(LPRDA) polypeptide, which was shown to possess antivirulence activity against S.aureus. the authors  established the framework for estimation of the key interactions and subsequent modification of LPRDA, targeted at non-peptide molecules, with better drug-like properties than the original polypeptide. The authors used the docking protocol to establish its applicability to the LPRDA-SrtA complex prediction. then, molecular dynamics studies were carried out to refine the geometries and estimate the stability of the complexes, predicted by docking. Finally, the hypothesis about SrtA binding is formulated.  Therefore, this article with its current form needs a major revision to qualify for publication in the Journal. please check the attached pdf file.

Docking study very lack

  • The manuscript needs rechecking for typos and grammar errors.
  • The authors ignore the morphology of LPRDA, as calculated HOMO, LUMO, and MEP using  DFT calculations and compare them with the X-RAY data.
  • The author’s carried out the Docking simulation, but this study is very lacking. The fit induces molecular docking was performed but not a validation of the molecular docking for the reported compound. The authors should mention the binding energy and rmsd at the molecular docking.
  • The DOCKING is not validated and other biochemical parameters are not carried out, as the ligand efficacy should calculate in molecular docking.
  • Author’s should calculate H-interaction energy also length for these bonds inactive sites to support these postulations.
  • The authors did not perform bioactivity checkups for validation purposes. Author’s should calculate physicochemical properties as %ABS, Mutagenic, Tumorigenic, Reproductive Effective, Irritant…etc.
  • The time (10 ns.) for MD calculations is not sufficient, the authors should discussed at 250ns. to evaluate the calculations.
  • The MD   don not mentioned as an ordinary method, and the authors don’t mention the relation between; time and  RMSD, residues, and  RMFS, …… etc.
  • The conclusion part should be rewritten in more detail was modified to be more comprehensive.

Author Response

Q#1: The manuscript needs rechecking for typos and grammar errors.

R#1: We undertook additional checking for typos and grammar errors.

Q#2: The authors ignore the morphology of LPRDA, as calculated HOMO, LUMO, and MEP using  DFT calculations and compare them with the X-RAY data.

R#2: LPRDA is a rather regular polipeptide in terms of quantum chemistry considerations. There is no any extensive conjugated groups, so the properties of the polypeptide should be pretty well described in terms of the properties of the consituiting aminoacids with little synergy (near additively). Thus we did not conduct such kid of studies. The experimental LPRDA structure is not known to the best of our knowledge. Even if it was known, it would not give the comprehensive understanding, because LPRDA contains ca 20 readily rotatable bonds and plenty of options to form intra- and intermolecular hydrogen bonds and salt bridges with different conformations and almost no penalty (due to high conformational flexibility). Another difficulty arising from the above is that the atomic charges used for AutoDock docking and MD should well reflect interactions for all feasible conformations. If charges are taken from a single conformation where several groups interact with each other strongly, the same charges would poorly describe the electrostatic interactions for the other conformations. For this reason we deliberately refused to derive AM1-BCC or RESP charges, which possess the above described deficiency (for this case). Instead we advise the use of topologically symmetric MMFF94 charges, which had been fitted to the same RHF/6-31G* level of MEP, which was used to parameterize GAFF/AMBER. Moreover, being atomic charge method developers for a long time, we are sure that the straight use of RESP or AM1-BCC for this polypeptide is WRONG for the above reasons. At least it's unfeasible that it leads to the correct results of both docking and MD.

Q#3: The author’s carried out the Docking simulation, but this study is very lacking. The fit induces molecular docking was performed but not a validation of the molecular docking for the reported compound. The authors should mention the binding energy and rmsd at the molecular docking.

R#3: The binding energies are provided and analyzed for each of the docking versions, both for AutoDock Vina and AutoDock. The energies obtained are in reasonable agreement with those from the work by Wang et al. (it is reflected in the text). Since no experimental structure for LPRDA is known, we cannot calculate RMSD with respect to it.

Q#4: The DOCKING is not validated and other biochemical parameters are not carried out, as the ligand efficacy should calculate in molecular docking.

R#4: Due to the lack of the experimental data for LPRDA/SrtA we cannot validate the docking directly. Even a meaningful validation using the innate ligand from PDB:1T2W is not feasible for the reasons discussed in the manuscript. First, the position from PDB:1T2W is hardly reasonable for the analysis, which is reflected in the section devoted to PDB complexes analysis. Second, Kappel et al [10.1002/pro.2168] arrived at the same conclusion while analyzing this complex. Thus, since the complex geometry is itself under question, its use for docking validation is not warranted. Ligand efficiency (LE) values are useful metrics for ligand optimization. We added the LE values for the complexes, reported in the manuscript with the docking estimated free energy of binding (via scoring functions).

Q#5: Author’s should calculate H-interaction energy also length for these bonds inactive sites to support these postulations.

R#5: Yes, we were planning to conduct such an analysis as well as MM/GBSA analysis and per residue energy decomposition in order to characterize the binding mode or several alternative binding modes. Our actual finding is, contrary to the work of Wang et al, it's hard to single out a certain mode per se. Yes, in agreement with the work of Kappel et al [10.1002/pro.2168] some modes are more frequent than the others, but still they used cluster analysis to separate those "modes". LPRDA from our work has two additional formal charges (D instead of T, and R instead of A) compared to LPATG polypeptide from the work of Kappel. Thus, the relative mobility of LPRDA could be expected to be even larger. By this work we strive to point out the specifics of the system LPRDA/SrtA we detected using meaningful and careful molecular modeling approaches. We believe they are useful for other researchers, who would also start from the standard protocols, in order to not be surprised by the results.

To partially support our hypothesis we included the average and standard deviation of the number of H-bonds along the MD trajectories.

Q#6: The authors did not perform bioactivity checkups for validation purposes. Author’s should calculate physicochemical properties as %ABS, Mutagenic, Tumorigenic, Reproductive Effective, Irritant…etc.

R#6: In the current work we do not present LPRDA per se. It was done by the work of Wang et al. We tried to establish the modeling protocol to reproduce the modeling results of Wang and start to use this protocol to develop our peptidomimetics - more potent and bioavailable analogs of LPRDA.

Q#7: The time (10 ns.) for MD calculations is not sufficient, the authors should discussed at 250ns. to evaluate the calculations.

R#7: As already mentioned in the manuscript, 10 ns MD time is not sufficient to definitely check the stability of the complexes, but it's sufficient to uncover unfavorable intermolecular interactions, which could have resulted from the docking. Moreover, we surprisingly discovered that the same 10 ns of MD in conjunctions with LPRDA complexed resulted in the enhanced sampling of the b6/b7 and b7/b8 loops, which usually take far longer. Thus the system was able to get rid of the most unfavorable interactions and initially fit (via b6/b7 and b7/b8 loops movement) to the better modes. Several modes were continued just for 100 ns to ensure the initial findings are correct. It was the case.

Q#8: The MD   don not mentioned as an ordinary method, and the authors don’t mention the relation between; time and  RMSD, residues, and  RMFS, …… etc.

R#8: We provided the corresponding RMSD and RMSF values in the SI.

Q#9: The conclusion part should be rewritten in more detail was modified to be more comprehensive.

R#9: Yes, we agree the main findings should be more comprehensively described. We did it in the edited version of the manuscript.

Reviewer 3 Report

The work describes a theoretical study aiming at establishing the framework for estimation of the key interactions and subsequent modification of LPRDA, targeted at non-peptide molecules, with better drug-like properties than the original polypeptide.

This is a well-prepared manuscript with good problem outlining, clearly stated objectives and approach, justified using existing literature. The docking and support initial docking calcs with MD methods are appropriate. Results presentation is plausible.

Overall, a great start to tackling SrtA inhibition using LDPRA.

This reviewer recommend acceptance after minor revision:

Abstract does not indicate the main findings. And citations should not be used.

Figures are informative and well presented. But there is large blank space amonst the manels. Please compact them by reducing blank space.

Table 1 should give the sources.

SI is missing, or inaccessible. Are all calcs in the study displayed in the manuscript? If no,

some MD data would also be useful. 10 ns is a good length of time but authors should provide some RMSD data at least so we can see that it equilibrated (can easily generate this in VMD).

Minor English usage problem needs to be rectified, e.g. L10, “…to combat the

virulence of that clinically important bacteria”; L793, “using as a base the work of

Wang et al.”; L820, “that” can not be used after “,”  There are more.

Author Response

Q#1: Abstract does not indicate the main findings. And citations should not be used.

R#1: Thank you for the advice. Done.

Q#2: Figures are informative and well presented. But there is a large blank space amongst the panels. Please compact them by reducing blank space.

R#2: Done, thanks.

Q#3: Table 1 should give the sources.

R#3: The sources were added to the table.

Q#4: SI is missing, or inaccessible. Are all calcs in the study displayed in the manuscript? If not, some MD data would also be useful.

R#4: We added the SI with RMSD and RMSF values for ligand and receptor for both 10 ns and 100 ns MD runs.

Q#5: 10 ns is a good length of time but authors should provide some RMSD data at least so we can see that it equilibrated (can easily generate this in VMD).

R#5: RMSD analysis was added to the SI.

Q#6: Minor English usage problem needs to be rectified, e.g. L10, “…to combat the virulence of that clinically important bacteria”; L793, “using as a base the work of Wang et al.”; L820, “that” can not be used after “,”  There are more.

R#6: We did our best to correct the wordings in English.

Round 2

Reviewer 1 Report

The authors have answered all my questions.